# Role of Gut Microbial Metabolites in the Pathogenesis of Primary Liver Cancers

**DOI:** 10.3390/nu16142372

**Published:** 2024-07-22

**Authors:** Maria Pallozzi, Valeria De Gaetano, Natalia Di Tommaso, Lucia Cerrito, Francesco Santopaolo, Leonardo Stella, Antonio Gasbarrini, Francesca Romana Ponziani

**Affiliations:** 1Liver Unit, Centro Malattie dell’Apparato Digerente (CEMAD), Medicina Interna e Gastroenterologia, Fondazione Policlinico Universitario Gemelli Istituto di Ricovero e Cura a Carattere Scientifico, IRCCS, 00168 Rome, Italy; mariapallozziucsc@gmail.com (M.P.); valeria.degaetano01@gmail.com (V.D.G.); ditommasonatalia@gmail.com (N.D.T.); lucia.cerrito@policlinicogemelli.it (L.C.); santopaolofrancesco@gmail.com (F.S.); leonardo.dr.stella@gmail.com (L.S.); antonio.gasbarrini@unicatt.it (A.G.); 2Dipartimento di Medicina e Chirurgia Traslazionale, Università Cattolica del Sacro Cuore, 00168 Rome, Italy

**Keywords:** inflammation, gut–liver axis, gut microbiota, hepatocellular carcinoma, cholangiocarcinoma, immunotherapy

## Abstract

Hepatobiliary malignancies, which include hepatocellular carcinoma (HCC) and cholangiocarcinoma (CCA), are the sixth most common cancers and the third leading cause of cancer-related death worldwide. Hepatic carcinogenesis is highly stimulated by chronic inflammation, defined as fibrosis deposition, and an aberrant imbalance between liver necrosis and nodular regeneration. In this context, the gut–liver axis and gut microbiota have demonstrated a critical role in the pathogenesis of HCC, as dysbiosis and altered intestinal permeability promote bacterial translocation, leading to chronic liver inflammation and tumorigenesis through several pathways. A few data exist on the role of the gut microbiota or bacteria resident in the biliary tract in the pathogenesis of CCA, and some microbial metabolites, such as choline and bile acids, seem to show an association. In this review, we analyze the impact of the gut microbiota and its metabolites on HCC and CCA development and the role of gut dysbiosis as a biomarker of hepatobiliary cancer risk and of response during anti-tumor therapy. We also discuss the future application of gut microbiota in hepatobiliary cancer management.

## 1. Introduction

Worldwide, hepatobiliary malignancies are the sixth most common cancer and the third leading cause of cancer death [1]. Hepatobiliary cancers include hepatocellular carcinoma (HCC), which is the fourth leading cause of cancer-related death and the main cause of death in patients with compensated chronic liver disease [2]. Cholangiocarcinoma (CCA) is the second most frequent primary liver cancer following HCC, and it accounts for up to 15% of primary hepatic tumors [3]. The gut microbiota is a key regulator of host homeostasis. In recent years, the disruption of the gut–liver axis and gut dysbiosis have been shown to play a critical role in promoting liver diseases.

The gut microbiota is made by the complex collection of various microorganisms (bacteria, archaea, eukarya, and viruses) that colonize the intestine and cooperate in multiple functions, such as host nutrient metabolism, xenobiotic and drug metabolism, immune modulation, and protection against pathogens through the maintenance of the structural integrity of the intestinal barrier [4]. The advances in anaerobic culture techniques and the development of culture-independent approaches (i.e., gene sequencing) have led to a better understanding of the composition and function of the “healthy” gut microbiota [5], which is mainly composed of seven phyla (Firmicutes, Bacteroidetes, Proteobacteria, Actinobacteria, Verrucomicrobia, Fusobacteria, and Cyanobacteria, with a predominance of Firmicutes and Proteobacteria) [6,7]. In homeostatic conditions, Firmicutes and Bacteroidetes represent 90% of the bacteria resident in the gastrointestinal tract, followed by Actinobacteria. From the oral cavity to the rectum, the number, density and diversity of microbiota varies. This balance is highly influenced by multiple factors, and it is disrupted during inflammation, infection or metabolic-associated conditions, such as alcohol consumption, unhealthy diet, obesity, and diabetes [8]. A high grade of gut dysbiosis has also been reported in patients with cancer, suggesting a causative role for gut dysbiosis in oncogenesis through multiple pathways [9,10]. A strict association between *Helicobacter pylori* infection and gastric cancer has been extensively demonstrated [11], and an increase in the abundance of Bacteroidetes, Firmicutes and Proteobacteria has been associated with esophageal cancer development [12]. Colorectal cancer occurrence is also influenced by the overgrowth of *Bacteroides fragilis*, *Clostridium septicum*, *Escherichia coli* [13], *Enterococcus faecalis* [14], *Fusobacterium* spp. and *Streptococcus bovis*, as described by many studies [15,16]. The gut microbiota can also influence tumorigenesis through the production of metabolites that may originate directly from bacteria or the bacterial transformation of dietary compounds or host-produced molecules; these include short-chain fatty acids (SCFAs), branched-chain amino acids (BCAAs), trimethylamine N-oxide (TMAO) and bile acids (BAs) [17]. In this review, we describe the role of the gut microbiota and its metabolites in the pathogenesis of HCC and CCA, and we also report the evidence on the predictive and prognostic importance of gut dysbiosis as a novel biomarker.

### 1.1. The Gut-Biliary-Liver Axis: Between Inflammation and Immunosuppression

One first role of gut microbiota in promoting HCC is linked to impaired functioning of the gut–liver axis. The gut and the liver are functionally and anatomically linked by the gut–liver axis, which includes the intestinal barrier and consists of vascular and lymphatic structures, as well as molecular and immune pathways that are deeply influenced by the gut microbiota [18,19]. The intestinal barrier is made up of multiple layers (the epithelium, the lamina propria, and the endothelium) that influence each other and present specific functions for the preservation of host homeostasis [20]. In healthy conditions, the intestinal barrier prevents the translocation of the gut bacteria and their products into systemic circulation. Any perturbation that produces an imbalance between the intestinal barrier and the gut microbiota affects the gut–liver axis and acts as a trigger for several liver diseases, including liver cirrhosis and HCC [21]. It is not surprising that gut dysbiosis represents the main driver in the gut–liver axis derangement, as it remarkably influences the intestinal barrier integrity and, as a consequence, pathological bacterial translocation (BT) is favored [22]. The presence of dysbiosis increases the concentration of enterotoxins in the gut, which stimulates the inflammatory response and increases intestinal barrier leakiness through the disruption of epithelial tight junctions (TJs). It is reported that elevated levels of zonula occludens 1 (ZO-1) are associated with increased intestinal permeability and disease severity in HCC [23]. Bacterial products such as the endotoxin lipopolysaccharides (LPSs) are also released in systemic circulation, leading to low-grade chronic inflammation [24,25]. LPSs are involved in several pathways that converge on HCC; interestingly, patients with liver cirrhosis and those with HCC show higher levels of circulating LPSs, according to animal models of carcinogen-induced hepatocarcinogenesis [26]. LPSs directly damage TJs through the activation of the Toll-Like Receptor (TLR) 4-Myeloid differentiation primary response 88 (MyD88) pathway, which activates myosin light chain kinase and enhances intestinal permeability. Moreover, through this same pathway, LPS acts as an inflammatory trigger for the liver, stimulating Kupffer cells to release pro-inflammatory cytokines such as interleukin (IL)-1 beta, IL-6, and tumor necrosis factor (TNF) alpha via the activation of the nuclear factor kappa B (NF-kB) cascade [27,28,29]. Also, Gram-positive bacteria influence systemic inflammation and carcinogenesis through the release of lipoteichoic acid (LTA), which acts on TLR-2 and converges eventually on Myd88. Indeed, in an obesity-driven murine model of HCC, the activation of TLR-2-Cyclooxygenase-2 (COX-2) [30] by LTA produced by Gram-positive bacteria was proven to favor hepatocarcinogenesis through two different mechanisms: the senescence of hepatic stellate cells (HSCs) and the modification of bile acid composition [31,32,33]. Moreover, gut dysbiosis and gut-derived products stimulate HSCs to express the senescence-associated secretory phenotype (SASP), reprogramming their transcriptome towards the production of inflammatory cytokines, such as IL-1 beta, as observed in mice models of non-fibrotic HCC after the establishment of diet-induced gut dysbiosis [32]. To analyze the possible contribution of HSC senescence and HCC development, Yoshimoto and colleagues used a mice model knock-out for the gene of IL-1 beta, a cytokine associated with both cancerogenesis and tumor suppression since it acts on the tumor microenvironment (TME) [32,34,35], and observed a parallel reduction in activated HSCs and the number of liver tumors [32,35]. Concerning the LTA/TLR-2/COX-2 pathway, it promotes the activation of prostaglandin E2 (PGE2) in HSCs, which is associated with a reduced anti-tumor response in the TME [36]. Moreover, TLR-4 triggers proliferative and antiapoptotic signals in resident hepatocytes of non-myeloid origin that are involved in cell growth and angiogenesis through the activation of STAT3 and its downstream genes cyclin D1, Bcl-2, c-myc, and IL-10 [25,37]. The activation of janus kinase/mitogen-activated protein kinase (JNK/MAPK) signaling by LPSs in HCC cells via TLR-4 results in enhanced invasive capacity and an epithelial–mesenchymal transition [38]. The interplay between low-grade chronic inflammation and gut dysbiosis also impacts immune system efficacy [39,40,41]. This has been observed in colorectal cancer, where gut dysbiosis, represented by a higher abundance of *Fusobacterium nucleatum*, has been associated with the impairment of natural killer (NK) cell expression and activity and a reduction in cluster differentiation (CD)-3+ T cells. These changes resulted from the direct interaction between *F*. *nucleatum* and the immune checkpoint “T cell immunoglobulin and immunoreceptor tyrosine-based inhibitory motif” (TIGIT), which is expressed on the surfaces of immune cells to downregulate their activity [42]. Bacterial translocation is a key player in this mechanism, as high circulating levels of bacterial extracts from patients with HCC injected in healthy patients induce a peculiar pattern of lymphocytes in the peripheral blood mononuclear cells, which are enriched in regulatory T cells and poor in cytotoxic CD-8 T cells [37,43,44]. How dysbiosis influences the anti-tumor response is not completely defined. Bacteria can stimulate immune cells differently. Many examples of the regulatory effect of the gut microbiota on immune cells through the upregulation of immune checkpoint molecules are reported in HCC patients: the overexpression of Programmed Death 1 (PD-1), Programmed Death Ligand 1 (PD-L1) and Cytotoxic T-Lymphocyte Antigen 4 (CTLA-4) leads to an increase in regulatory T cells, while CD 4+ T cells, CD 8+ T cells and NK cells are downregulated [45]. In patients with HCC, *Bacteroidetes* and *E*. *coli* were associated with higher levels of these antagonists of the immune response, resulting in a reduced efficacy of immunotherapy [45,46,47]. Gut-derived metabolites may have a role in this interaction; in particular, SCFAs inhibit the histone deacetylase, which limits the expression of PD-1 and results in its upregulation, but their role has not been clarified yet [48,49,50,51]. Also, primary bile acids promote the accumulation of CXC receptor 6+ NK T cells in the liver, which favors tumor inhibition, whereas secondary bile acids have opposite effects [52]. Changes in the gut–liver axis and gut dysbiosis are considered key players in the development of chronic disease of the biliary tract. Preclinical studies demonstrated that in healthy conditions, bile ducts are sterile; intestinal barrier disruption causes the translocation of bacteria and LPSs in the portal circulation and in the biliary tract. In this condition, gut microbiota and its products are able to activate TLRs exposed by cholangiocytes. In response to chronic TLR activation, cholangiocytes produce a variety of inflammatory cytokines, such as IL-1, IL-6, IL-8, TNF, transforming growth factor (TGF), and interferon gamma (IFN gamma), leading to an aberrant reaction that results in fibrosis, bile cell proliferation, and the generation of an immunosuppressive milieu [53,54]. In particular, the upregulation of myeloid-derived suppressor cells (MDSCs) causes the inhibition of cytotoxic T-lymphocytes and influences tumor replication, intravascular invasion, and metastasis. Hepatocytes contribute to this process by secreting CXC motif chemokine ligand 1 (CXCL1), which interacts with the CXC chemokine receptor on the aforementioned myeloid cells [55]. Thus, the chronic activation of TLR-4 and the upregulation of its gene correlates with CCA occurrence and progression [56,57]. Another possible mechanism involved in cholangiocarcinogenesis is molecular mimicry, observed in autoimmune conditions such as primary biliary cholangitis; this consists of the aberrant production of autoantibodies that are very similar, have bacterial and virus epitopes, and are able to activate effector T cells. According to this hypothesis, autoantibodies participate in the promotion and maintenance of liver inflammation and tumorigenesis [58,59,60].

### 1.2. Gut Dysbiosis and Hepatobiliary Carcinogenesis

Liver cirrhosis is characterized by a profound derangement of the gut microbiota composition. In particular, while bacterial alpha diversity and the abundance of Lactobacilli and Bifidobacteria is reduced, a marked increase in Enterobacteriaceae, Enterococci, Bacteroides, and Ruminococcaceae is observed [61,62,63]. Aerobic Gram-negative bacteria, specifically *E*. *coli* and other Enterobacteriaceae, have been identified as the most frequently involved in bacterial translocation and the major source of serum endotoxin [64]. On the other hand, anaerobic bacteria such as Bifidobacteria and Lactobacilli limit the overgrowth of potentially invasive microorganisms and exert ant-inflammatory and homeostatic properties [65]. In patients with HCC, an abundance of *E*. *coli* [66], *Actinobacteria* and other bacteria that produce LPSs has been observed with a reduction in butyrate-producing ones [62]; moreover, stools from patients with non-viral HCC harbored more pro-inflammatory bacteria, such as *Escherichia*, *Shigella* and *Enterococcus*, and less *Faecalibacterium*, *Ruminococcus, and Ruminoclostridium* spp., which instead produce anti-inflammatory compounds such as SCFAs [67]. Gut dysbiosis is a key player in the development of HCC in patients with NAFLD. In overweight patients, bacterial overgrowth syndrome is observed, and this is associated with higher levels of LPS released, with consequent hyperstimulation of inflammation. Inflammation also stimulates cell proliferation and influences immune cell downregulation. Intestinal permeability is another key regulator of this process, with a high release of LPSs into circulation, which is associated with the progression from NAFLD to NASH [32,68]. Modifications in gut microbiota have been associated with NAFLD, with significant changes occurring due to the evolution of liver fibrosis. In particular, it is observed that there is a loss of bacterial diversity associated with the depletion of *Lactobacillus* and *Bifidobacterium* spp. in association with the increase in *Ruminococcus* and *Escherichia* spp. when fibrosis is detected. In this condition, changes in gut-derived metabolites are mandatory, in particular, BA changes are crucial in the development of obesity-associated HCC [69,70]. In obesity, higher levels of DCA are observed, and this stimulates the expression of the SASP in hepatic stellate cells, which release inflammatory cytokines that raise the risk of HCC. Similar effects are observed in HCC that occurred in patients with non-alcoholic steatohepatitis. The activation of TLR in NASH models by LPS or bacterial DNA also activates HSCs, leading to fibrosis [69,70]. In patients with non-alcoholic steatohepatitis (NASH)-related cirrhosis and HCC, *Bacteroides* and *Ruminococcaceae* are increased, with a relative reduction in *Bifidobacterium* and *Akkermansia* [63]. Ren et al. analyzed fecal samples of 75 patients with cirrhosis and early HCC, 40 cirrhotic patients and 75 healthy controls from a heterogeneous Chinese population to identify possible genomic biomarkers of HCC. Patients with early HCC showed a greater microbial diversity and an increase in Actinobacteria phylum and *Gemmiger*, *Parabacteroides* and *Paraprevotella* genera in contrast to cirrhotic controls; moreover, butyrate-producing bacteria were decreased while LPS-producing bacteria were increased [62]. Also, an increased fecal abundance of Gram-negative *E*. *coli* has been found in the stool of cirrhotic patients with HCC, and this may lead to enhanced intestinal and hepatic inflammation mediated by the LPS-TLR4-NF-kB pathway. This was recapitulated in a mouse model of diethylnitrosamine (DEN) plus carbon tetrachloride (CCl4)-induced HCC, as animals with nonfunctional TLR-4 (TLR4-mut) had a significant reduction in tumor number and size but no difference in tumor incidence when compared to TLR-4 wild-type mice exposed to the same oncogenic toxins [37]. Gut sterilization obtained by the administration of antibiotics decreased LPS serum levels and was associated with an effect on tumor size and number similar to TLR-4 mutation, further confirming that gut microbiota-driven inflammation favors HCC progression [37]. Recent studies have also focused on the role of intratumor microbiota [65,66]. Li et al. [67] analyzed tissues derived from 29 patients with HBV-related HCC using next-generation sequencing (NGS) and discovered a lower bacterial diversity in tumor samples compared to non-tumor ones. HCC tissues were also divided into two subtypes: one with bacteria predominance (Bacteria-T) and the other with virus predominance (Virus-T). In Bacteria-T, *E*. *coli*, *Shigella dysenteriae*, *Babesia bigemina* and *Pannonibacter phragmitetus* were enriched, while HBV was prevalent in Virus-T. Bacteria-T HCC was characterized by larger size and capsular invasion and a higher prevalence of M2 macrophages in the TME compared to Virus-T [71,72]. Compared to patients with HCC or liver cirrhosis, the gut microbiota of patients affected by intrahepatic CCA (iCCA) is enriched in four genera (*Lactobacillus*, *Actinomyces*, *Peptostreptococcaceae* and *Alloscardovia)*, and Ruminococcaceae abundance was positively associated with vascular invasion [72]. In vitro experiments showed that the most described intratumor microbiome was composed of Burkholderiales, Pseudomonadales, Xanthomonadales, Bacillales and Clostridiales. Moreover, Paraburkholderia fungorum was associated with CA19-9 levels and located in the paracancerous tissues [73]. Other bacteria involved in cholangiocarcinogenesis are *H*. *pylori*, *H*. *hepaticus* and *Opisthorchis viverrini*, and the most abundant genera identified in the biliary microbiota were *Enterococcus*, *Streptococcus*, *Bacteroides*, *Klebsiella* and *Pyramidobacter* spp. [74]. Figure 1 summarizes the relationship between the impairment of gut–liver axis homeostasis and hepatic tumorigenesis. Table 1 synthesizes the main features of gut microbiota composition in primary liver cancers.

## 2. Gut Metabolome

While the “direct” effect of bacterial products on inflammation and tumorigenesis has been well established, understanding the role of gut metabolites in the pathogenesis of primary hepatobiliary cancers is a page yet to be written. The gut microbiota acts as a bioreactor of metabolic functions through the enzymatic transformation of endogenous and exogenous substances into its metabolites, influencing host homeostasis. Gut dysbiosis alters the levels of metabolites, including those involved in carcinogenesis, causing dramatic consequences [75,76]. Data on the role of gut-derived metabolites in carcinogenesis are accumulating; here, we describe the main results on the role of BAs, SCFAs, ethanol, BCAAs and indole-derived products.

### 2.1. Bile Acids

BAs are steroid derivatives of amphipathic molecules produced in the liver and are transformed by the gut microbiota into signaling agents. Despite the fact that the production of primary BAs is a prerogative of the host, secondary BAs are derived from an active transformation mediated by the gut microbiota through a process involving the deconjugation, dehydroxylation and dehydrogenation of primary BAs mediated by colonic bacteria with bile salt hydroxylate activity [77]. As a consequence, BAs may be assimilated into gut-derived metabolites with significant signaling activity. In healthy conditions, primary BAs are transported with the bile in the intestine, where they are converted into secondary BAs. BAs participate in several functions, such as the elimination of cholesterol and catabolites, the emulsification of fat-soluble vitamins, the regulation of intestinal motility, and the maintenance of balance in the small intestinal and biliary tract microbiota [31,33,77]. In the colon, the gut microbiota participates in debinding water from BAs as well as in their deconjugation, dihydroxylation, and dehydrogenation. Regarding deconjugation, *Bacteroides*, *Clostridium*, *Lactobacillus*, *Bifidobacterium*, *Listeria*, and *Escherichia* spp. are involved in this process, showing bile salt hydrolase (BSH) activity [78], and unconjugated primary BAs are further dehydrogenated and 7α-dehydroxylated mainly by *Clostridium* spp. forming secondary BAs. In the presence of chronic inflammation, gut dysbiosis leads to an imbalance in the intrahepatic levels of BAs since it negatively influences farnesoid X receptor (FXR) expression, a key regulator of BA metabolism and signaling, resulting in hepatotoxicity [78] through the modulation of hepatic transporters. This mechanism negatively influences farnesoid X receptor (FXR) expression, leading to hepatotoxicity. Indeed, excessive levels of BAs damage the plasma membrane, activating the protein kinase C/MAPK/NFkB pathway that upregulates inflammatory cytokines such as TNF alpha, IL-1 beta, and IL-6. These cytokines counteract the apoptotic processes through the JAK-STAT3 and phosphatidylinositol 3-kinase (PI3K) pathways, favoring cell immortalization [79]. Hydrophobic secondary BAs, including cholic acid (CA), glycocholic acid (GCA), lithocholic acid (LCA), chenodeoxycholic acid (CDCA), and deoxycholic acid (DCA), are the main ones responsible for hepatocyte death [80]. Their action appears to be mediated by the activation of TNF-related apoptosis-inducing ligand receptor (TRAILR) and Fas death receptor signaling pathways [81]. Another mechanism that leads to tumorigenesis is the exaggerated activation of apoptotic pathways directly or indirectly mediated by BAs; indeed, BAs mediate the transfer of Fas-containing vesicles to hepatocyte membranes. The interaction between Fas and a death-inducing signaling complex composed of the Fas-associated death domain and procaspases 8 and 10 activates apoptosis; specifically, procaspases are activated into caspases 8 and 10, and Bax is translocated into mitochondria. Other mechanisms include cytochrome c release and caspase 9 activation [82]. Moreover, the release of cytochrome c can occur through the direct action of BAs in Bax translocation and through BA-induced reactive oxygen species (ROS) production [79]. Finally, BAs (DCA and G-CDCA) induce endoplasmic reticulum (ER) stress, resulting in the release of Ca2+ in the cytoplasm and the further promotion of oxidative stress [83]. The stimulation of proliferative pathways through the activation of an inflammatory cascade that leads to the upregulation of gene signaling is another common mechanism involved in tumorigenesis mediated by BAs: BAs upregulate Early growth response-1 (Egr-1), which is required for the development of liver inflammation during cholestasis, via MAPK signaling [80] directly or via previous activation of FXR, which heterodimerizes with RXR and modulates gene expression [84]. Furthermore, CDCA and DCA activate epithelial growth factor receptor (EGFR) and upregulate Egr-1, causing the production of vascular endothelial cell adhesion molecule-1 (VCAM-1), IL-1β, and IL-10 in hepatocytes [85]. Moreover, damage to the plasmatic membrane by toxic BAs triggers protein kinase C (PKC), which, in turn, activates the MAPK cascade, leading to the activation of NF-kB and the production of IL-6, IL-1 beta, and TNF alpha [79,86]. FXR is the most important nuclear receptor for BAs involved in the process of hepatocarcinogenesis. Another mechanism of hepatocarcinogenesis influenced by BAs is related to the inhibition of FXR signaling. FXR inhibits HCC development through the modulation of several metabolic pathways involving BAs, glucose, and lipids, the suppression of liver inflammation, the promotion of tissue repair after liver injury, the expression of partial tumor suppressor genes, and the inhibition of transcription of multiple oncogenes [87,88]. When binding to caspase 8, FXR prevents the activation and conduction of apoptotic signals in a ligand-independent pathway. During liver injury, in response to high levels of TRAIL, TNFα, and Fas ligand, FXR expression reduces, suggesting that the decrease in FXR in hepatocytes precedes apoptosis [81]. BAs can downregulate FXR through the reduction of sirtuin 1 (SIRT1), a transcriptional regulator of FXR expression in hepatocytes [89]. This results in sustained activation of the Wnt/β-catenin pathway and increased risk of hepatocarcinogenesis [90]. A diet enriched in lipids increases the levels of the gut microbiota-derived DCA, which is known to cause DNA damage. In hepatic stellate cells, DCA induces the SASP, turning in the secretion of inflammatory and tumor-promoting factors in the liver [32] and the overexpression of COX-2 and its downstream inflammatory cascade, together with TLR-2-mediated signaling [6,25,52]. BAs also influence the immune response and tolerance against tumors. Commensal bacteria are important regulators of anti-tumor immunity, and gut microbiome-mediated primary-to-secondary BA conversion can regulate the accumulation of NK T cells by CXCL16 expression in liver sinusoidal endothelial cells [91]. Primary BAs such as CDCA and toxic CA increase CXCL16 expression, which binds to its receptor CXCR6 on NKT, thereby increasing IFN gamma production and inhibiting tumor growth [6]. Lithocholic acid was also reported to influence the balance between Th17 and T regulatory cells by acting on the retinoic acid-associated orphan receptors gamma T [91]. As gut-derived metabolites may modulate the risk of HCC development acting on modulatory signaling pathways, several drugs, such as FXR agonists, are under evaluation. The inhibition of FXR functions is associated with impaired cell proliferation and DNA damage, while its activation exerts a protective role on tumorigenesis [92,93]; in a mouse model, the administration of FXR agonists (GSK2324) was able to inhibit lipogenesis, the intestinal uptake of lipids and their intrahepatic accumulation, modulating the levels of monounsaturated fatty acids, polyunsaturated fatty acids and BAs [94]. It is not known whether BA qualitative changes observed in CCA patients are a cause or a consequence of the presence of CCA since this tumor uses BA receptors and may influence BA metabolism. Preliminary evidence reports that in CCA patients, there is an increase in GCA, a decrease in secondary Bas, and changes in the ratio between glycoconjugated and tauroconjugated BAs. The tumor-associated production of IL-6 influences the expression of FXR and G Protein-coupled BA Receptor 1, known as TGR5 receptors. Patients with CCA and vascular invasion show higher levels of glycochenodeoxycholic acid (GCDCA), taurocholic acid (TCA), glycodeoxycholic acid (GDCA), taurodeoxycholic acid (TDCA) and tauroursodeoxycholic acid (TUDCA). On the other hand, in this study, the levels of IL-6, as well as those of CDCA, were lower in the group with vascular invasion than in patients without vascular invasion, which is discordant with previous observations on the association between BA accumulation, the increase in IL-6 and downregulation of FXR and cytochrome (CYP) 7A1, the key enzyme for BA synthesis. The existence of interrelationships among gut microbiota, BA profile, serum levels of cytokines, and iCCA patients’ outcomes deserves to be explored in larger studies [94]. Thus, BAs may damage the liver, stimulating carcinogenesis through the intrahepatic accumulation or the modulation of the main receptors of BAs, which leads to chronic inflammation, insufficient apoptotic processes, and aberrant cell proliferation.

### 2.2. Choline and TMAO

Choline is a product derived from the catabolism of dietary compounds such as meat and egg yolk. Gut microbiota promotes its metabolism into trimethylamine (TMA), mainly bacteria belonging to Firmicutes and Proteobacteria phyla (*Anaerococcus hydrogenalis*, *Clostridium asparagiforme*, *C*. *hathewayi*, *C*. *sporogenes*, *E*. *fergusonii*, *Proteus penneri*, *Providencia rettgeri* and *Edwardsiella tarda*) [95,96,97]. TMA obtained from this process reaches the portal circulation, and it is converted in the liver by flavin monooxygenase 3 into trimethylamine-N-oxide (TMAO). TMAO is associated with an increased risk of cardiovascular disease, as it promotes the synthesis of proinflammatory cytokines and the release of ROS from endothelial cells [98,99,100,101]. The role of choline in carcinogenesis is controversial. Dietary choline intake positively correlates with the occurrence of colorectal and prostatic cancers [102,103], even if a meta-analysis reported a protective role of this metabolite against tumorigenesis [104]. Moreover, in a recent study, high levels of dietary choline were able to reduce the risk of steatotic liver disease in humans; conversely, lower levels led to the accumulation of intrahepatic triglycerides, resulting in a higher risk of metabolic-associated disease, including HCC. [105,106,107]. Liu et al. demonstrated in a large Chinese population that patients with high plasma levels of TMAO and low choline showed a higher risk of developing liver cancers compared to individuals with high choline and low TMAO plasma levels, but the underlying mechanism has not been elucidated yet [108]. TMAO may stimulate the activation of mammalian target of rapamycin (mTOR) signaling through the upregulation of the periostin (POSTN) gene, as observed in a model of Hepa 1–6 cells and Huh7 cells exposed to TNF [109,110]. POSTN is a gene involved in tumorigenesis, the modulation of the tumor microenvironment, intravascular tumor dissemination and metastases [111]. Another trial conducted on 40 healthy men documented that fish consumption induced high TMAO production among individuals with low gut microbiota diversity and higher *Firmicutes* to *Bacteroidetes* ratio, recapitulating the essential contribution of gut microbiota in determining plasma levels of TMA and TMAO metabolites [112]. An interesting field of research is the functional interplay between BA metabolism and TMAO. The combination of TMAO and a high-fat diet (HFD) seems to have a synergistic effect on NAFLD development since TMAO could influence BA metabolism towards the increased synthesis of FXR-antagonists [113]; as we mentioned above [90], FXR inhibition is associated with spontaneous HCC development through the Wnt/β-catenin pathway. More studies are needed to understand whether dietary restrictions or gut microbiota manipulation toward lower TMA production could prevent HCC development and progression, especially in NAFLD-NASH-HCC. Preclinical studies showed that a choline-depleting diet may also stimulate CCA development. Rats treated with 2 acetylaminofluorene fed with a choline-devoid diet showed the occurrence of CCA after 6 months in 38% of cases, while a diet enriched with choline prevented tumorigenesis. Similar results were obtained using other tumor-inducing substances in association with a low-choline diet [114]. A low-choline and high-methionine diet for 6–12 months in mice caused the occurrence of primary liver tumors (both HCC and CCA). In all cases, osteopontin was upregulated and able to activate the beta-catenin signaling pathway and cause a decrease in endothelial cadherin [114]. No data about the effects of TMAO in CCA are reported in the literature. To summarize, low levels of choline and higher levels of TMAO are associated with a higher risk of liver cancer occurrence since choline influences intrahepatic fat deposition, leading to hepatotoxicity, and TMAO is involved in the stimulation of abnormal cell proliferation.

### 2.3. Indoles

Indoles are gut microbiota products derived from tryptophan metabolism that are produced by many bacterial species, mainly by *E*. *coli*. Indoles may cross the intestinal barrier and reach the liver, where they are converted in a two-step process into 3-hydroxyindole (indoxyl) and conjugated with sulfate in indoxyl sulfate. Moreover, molecules of indoxyl may dimerize to form the blue dye indigo [115,116]. These products act as aryl hydrocarbon nuclear receptors (AhRs). In the presence of indole derivatives (indoles), intestinal cells express AhRs, promoting immune cell development and intestinal barrier permeability. In the presence of inflammatory conditions such as chronic liver diseases, indole levels are reduced [117]. In a study by Krishnan et al., exposure to a high-fat diet significantly inhibited indole-3-acetate intestinal levels, resulting in the disruption of the intestinal barrier, overexpression of inflammatory cytokines, and inhibition of immune cells [118]. The anti-inflammatory effects of indoles may be due to the direct action on the host intestinal barrier and are partially attributed to changes in gut microbiota metabolism with no influence on its composition. Indoles also reduce fatty acid oxidation [119]. The oral supplementation of indoles reverts inflammatory processes [118] and may be evaluated in patients with chronic inflammatory diseases, including primary hepatobiliary cancers.

### 2.4. Short-Chain Fatty Acids

SCFAs are produced by the intestinal microbiota from the fermentation of indigestible carbohydrates. They are involved in the homeostasis of the intestinal epithelial barrier, and contribute to the maintenance of gut microbiota diversity [120,121,122,123,124]. Acetate, propionate, and butyrate are the main SCFAs produced in healthy individuals, with a molar ratio in the human colon of 60:20:20. Acetate derives from anaerobic bacteria such as *A*. *muciniphila* and *Bacteroides* spp; in normal conditions, it is involved in lipid synthesis, glucose metabolism, and immunity regulation [125]. Propionate is obtained during glycolysis through the succinate pathway and is commonly produced by the enzyme methylmanolyl-Coenzyme A (CoA) decarboxylase of Gram-negative bacteria such as *Akkermansia muciniphila* and *Roseburia inulinivorans*; it is involved in lipid metabolism, inhibits lipogenesis and favors liver gluconeogenesis [125,126]. Butyrate derives from Clostridium clusters IV and XIVa metabolism, and it is produced via glycolysis from the combination of two acetyl CoA molecules to form acetoacetyl CoA, which is reduced in butyryl CoA. Butyrate is finally obtained from the butyryl CoA acetate CoA transferase pathway or through the phospho butyrate and butyrate kinase pathway [125,127]. Butyrate is involved in the regulation of intestinal barrier permeability and intestinal epithelial cell metabolism, and it acts as an inhibitor of carcinogenesis in the colon [128]. SCFAs modulate human health via several pathways. First, they interact with GPR41, GPR43, and GPR109A, which activate the phospholipase C beta, the protein kinase A (PKA), and protein kinase C (PKC), leading to ERK activation. SCFAs also modulate the immune system, enhancing the production of macrophages, dendritic cells, and CD 4+ T cells. Moreover, butyrate inhibits inflammation in the colon, enhancing the production of T-regulatory cells and IL-10 through the GPR109A receptor [125,129]. In HCC patients, stool analysis revealed that SCFA-producing bacteria such as butyrate-producing *Lachnospira*, *Ruminococcus,* and *Butyricicoccus* are less represented [130]. Other studies showed an association of SCFAs with hepatocarcinogenesis [46]. Mice with elevated BAs and hyperbilirubinemia, when fed with inulin to induce SCFA production, showed higher liver inflammation, neutrophil influx and risk of developing HCC [131]. Enrichment in beneficial bacteria such as *Blautia* and *Lactobacillus* spp. may increase sodium butyrate levels with the aim of reducing inflammation and bile acids such as DCA [132,133]. SCFAs could also contribute to the development of an immunosuppressive microenvironment and HCC progression by promoting the production of IL-10 by microbiota antigen-specific T helper (Th) 1 cells or suppressing inflammatory macrophages in the lamina propria [134]. Indeed, some studies showed increased circulating levels of IL-10 and T-regulatory cells (T regs) after exposure to butyrate [135,136]. In a mouse model of hepatitis B virus-induced HCC, it was noted that SCFA administration reduced the number of dysplastic nodules and HCC in HBx transgenic mice, downregulating some genes critically involved in tumor progression, such as several growth factors, PI3K, Wnt, Vascular Endothelial Growth Factor (VEGF) and Ras [134]. It is recognized that SCFAs interfere with the cancer cell cycle through DNA epigenetic modifications [137] and histone deacetylase activity (HDAC) inhibition, which is a critical regulator of gene transcription that is often found to be dysregulated in cancer [138]. A recent study showed that Lactobacillus reuteri administration in HCC-recipient mice was positively associated with increased acetate plasma levels and contrasts tumor growth [139]. Also, L. reuteri could inhibit type 3 innate lymphoid cells (ILC3s), a subtype of tissue-resident lymphocytes, to produce IL-17A, a cytokine with a well-known role in favoring HCC development [139,140,141]. Researchers showed that acetate itself is able to downregulate ILC3 functions by blocking HDAC activity; thus, they documented that the combination of acetate plus anti-PD-1 therapy could enhance the immune response in HCC mice models receiving ICIs, decreasing IL-17A mRNA levels [139]. Moreover, supplementation with probiotics can be used for the same purpose; some bacteria, such as *Bifidobacterium pseudolongum*, produce acetate, which, through the interaction with GPR43 on hepatocytes, inhibits cell proliferation and induces apoptosis in a model of NAFLD-HCC mice [142]. Finally, SCFAs can activate the CD-8 T-cell response in patients receiving anti-PD-1 therapy by the release of IL-17 [143] and exert anti-tumor activity that inhibits CTLA-4 on dendritic cells and T cells. Butyrate, on the contrary, may stimulate an anergic phenotype that enhances T-regulatory cells. Butyrate metabolism is highly activated in patients with HCC compared with healthy individuals, resulting in reduced plasma levels of butyrate. Butyrate supplementation or inhibition of its catabolism significantly reduced liver tumorigenesis in a process that influences intracellular calcium homeostasis and the production of ROS. Moreover, higher levels of butyrate were associated with higher efficacy of tyrosine kinase inhibitor-based therapy; the administration of encapsulated pegylated butyrate and sorafenib nanoparticles prolonged drug retention time and optimized the anti-tumor response [144]. In vitro studies also demonstrated that CCA cells treated with butyrate showed a positive effect on cilia formation and acetylated tubulin levels. In HuCCT1 cell lines, migration, and mitosis were inhibited by the administration of butyrate. No effects were observed in non-tumor mature cells. Butyrate stimulates the effects of HDAC inhibition and reduces the levels of cyclin D1, vimentin, and other proteins involved in cell proliferation [145,146]. Thus, the interplay between SCFAs and HCC is complex and depends on multiple factors, including the degree of dysbiosis, the presence of other metabolites, such as BAs, and the conditioning of TME and the anti-tumor immune response [135].

### 2.5. Ethanol

In physiologic conditions, small amounts of endogenous ethanol are produced after the intake of alcohol-free food as metabolic intermediaries or products [147,148]. In the case of intestinal dysbiosis, the abundance of high alcohol producers such as *Klebsiella pneumoniae* increases, contributing to the pathogenesis of disorders such as the autobrewery syndrome or alcohol-mediated liver damage [148,149,150,151]. Also, other members of the gut microbiota, such as Proteobacteria, more specifically *E*. *coli* [149], and yeasts, such as *Saccaromyces cerevisiae* and Candida spp., play an important role in endogenous ethanol production [152]. The mechanisms of ethanol-induced carcinogenesis are closely related to its metabolism. Ethanol is converted into acetaldehyde by three pathways: (1) alcohol dehydrogenase (ADH), which leads to the formation of acetaldehyde and reduced nicotinamide adenine dinucleotide (NADH); (2) cytochrome P450 2E1 (CYP2E1), with the formation of reactive oxygen species; (3) catalase, but to a much lesser extent. Once formed, acetaldehyde is metabolized into acetate by acetaldehyde dehydrogenase (ALDH) [153,154]. Acetaldehyde is a well-known carcinogen since it interferes with DNA synthesis and repair [155,156,157]. Moreover, it causes structural and functional alterations to proteins to which it binds [158]. Levels of acetaldehyde do not always correlate with alcohol consumption, as the activity of ADH and ALDH is primarily responsible for the amount of acetaldehyde generated. The expression and activity of these enzymes depend on genetic factors [155]; however, the oral microbiome can also convert ethanol to acetaldehyde by oxidation, while further conversion in acetate is limited [159,160]. In the oral cavity, mainly Gram-positive aerobic bacteria and yeasts appeared associated with higher acetaldehyde production, with an increased risk of developing oral cavity malignancy [161]; moreover, compared to germ-free rats, conventional animals showed higher levels of intracolic acetaldehyde and a more pronounced mucosal injury and cellular hyper-regeneration [162]. Acetaldehyde does not appear to play a crucial role in hepatocarcinogenesis due to its effective hepatic metabolism and, consequently, low intrahepatic levels of this compound. Liver damage related to ethanol is rather driven by ROS production. As stated above, ethanol-mediated ROS formation involves CYP2E1; the chronic consumption of ethanol can induce and make this pathway predominant [163]. Moreover, NADH derived from the ADH pathway shuttles into mitochondria and generates electron leakage from the mitochondrial respiratory chain, forming ROS. ROS interact with lipids, generating lipid peroxidation products, such as trans-4-hydroxy-2-nonenal (4HNE), which can be converted to 2,3-epoxy-4-hydroxynonenal. This compound reacts with deoxyadenosine or deoxycytidine to form exocyclic etheno-DNA adducts such as 1,N6-ethenoadenine or 3,N4-ethenocytosine, causing DNA damage. To note, acetaldehyde and nitric oxide derived from inducible nitric oxide synthase induced by ethanol inhibit the DNA repair system; furthermore, acetaldehyde also increases the burden of ROS indirectly by injuring mitochondria, resulting in an inadequate reoxidation of the large quantities of nicotinamide adenine dinucleotide (NADH) that are produced through the alcohol dehydrogenase (ADH) reaction. As mitochondrial damage occurs, this cascade of events leading to apoptosis is initiated [164,165,166]. Thus, ethanol induces liver damage indirectly through the exaggerated production of ROS, which causes mitochondrial dysfunction or directly acting on DNA stability.

### 2.6. Branched-Chain Amino Acids

BCAAs are essential amino acids with an aliphatic chain that are involved in multiple anabolic processes, such as gluconeogenesis. They are degraded by the branched-chain aminotransferases (BCATs) 1 and 2 into branched-chain alpha ketoacids, which are fundamental for the skeletal muscle and the nervous system [167,168]. From the muscle, BCAAs may be converted by the branched-chain alpha ketoacid dehydrogenase to be released and utilized for gluconeogenesis. Free BCAAs stimulate the release of insulin and its receptor, the insulin receptor substrate (IRS1), leading to the activation of the PI3K/protein kinase B (AKT)/mTOR complex 1 (mTORC1); BCAAs may also stimulate the direct activation of the mTORC1 pathway, which is involved in cell proliferation, angiogenesis and apoptosis [167,168,169,170,171,172]. In the presence of gut dysbiosis, such as in HCC, BCAA levels increased not only in plasma but even in the tumor tissue and the surrounding area; interestingly, the downregulation of the catabolic enzymes of BCAAs is reported in tumor cells, leading to BCAA accumulation and mTORC1 activation. BCAA levels were also associated with tumor size and tumor number in a mice model of HCC. Another mechanism of HCC progression related to BCAAs is the production of mitochondrial distress through the accumulation of branched-chain ketoacids. The administration of BCAAs with diet seems to inhibit this process, suggesting that oral supplementation may enhance the anti-tumor response through the modulation of the gut microbiota; BCAAs may alter the gut microbiota composition, reducing serum levels. BCAA supplementation with diet increases the abundance of *Ruminococcus flavefaciens* and *Bifidobacterium* strains while reducing Proteobacteria, which are involved in several dysmetabolic conditions [164,165,166,167,168,169]. Intratumor degradation of valine, leucine, and isoleucine is observed in CCA due to the upregulation of BCAT 1 and BCAT 2 receptors. The levels of these amino acids are negatively associated with tumor prognosis [173]. In conclusion, dietary BCAAs may reduce the risk of tumorigenesis; on the contrary, gut dysbiosis and chronic inflammation through the modulation of enzymes involved in BCAA metabolism may cause intratumor accumulation of BCAAs, resulting in cell survival and the promotion of angiogenesis.

Table 2 and Figure 2 summarize the main gut-derived metabolites and their role in hepatocarcinogenesis. 

## 3. Discussion and Future Perspectives

Gut-derived metabolites are gatekeepers of liver homeostasis, inflammation, and tumorigenesis. Their quantification for the diagnosis or their use for the treatment of primary liver tumors has gained attention, thanks to the progress of metagenomics and proteomics techniques [6,174]. Liquid biopsy is based on these premises, employing the detection of a combination of products in biological material (stool, blood, urine) to identify the presence of a tumor and describe its metabolic features. A well-known application of liquid biopsy has been reported in colorectal cancer. Through the assessment of a panel of gut microbiome-associated serum metabolites derived from blood and fecal samples, a specific signature in colorectal cancer has been demonstrated, as well as the possibility of discriminating adenomas from tumors. This panel was superior to the carcinoembryonic antigen marker, with a high accuracy in detecting early-stage colorectal cancer [175]. Regarding gut-derived metabolites, the catabolism of BCAAs has been reported in several types of malignancies, leading to low levels in blood or saliva, and blood tryptophan levels were altered in patients with lung cancer and colorectal and pancreatic cancers [176]. Unfortunately, no significant similar studies are reported concerning HCC, except for low levels of butyrate in patients with early HCC. Similarly, the analysis of intratumor gut microbiota and its metabolites is another field of research for both HCC and CCA, promising to give novel information on the aggressiveness and resistance of tumor cells and the likelihood of response during systemic treatment [177]. Another promising field of the application of metabolomics regards HCC systemic therapy, specifically in the evaluation of the anti-tumor response [178]. Many studies are focused on the role of the gut microbiota as a key player in immunomodulation and as a biomarker of treatment response [179]. Changes in the gut microbiota have been associated with the development of early HCC. A high Firmicutes/Bacteroidetes ratio has been associated with a lower rate of response to immunotherapy, while the *Prevotella*/*Bacteroides* ratio has shown a positive association with the response in patients receiving Nivolumab. How gut microbiota may influence the anti-tumor response is not completely defined, and it is reported that different bacteria may stimulate immune cells differently; indeed, *Bacteroides fragilis* is able to activate CD-4 T cells to produce IFN gamma, while *Bifidobacterium* enhances the response to anti-PD-L1 and anti-PD-1, thus activating CD-8 T cells [180,181,182,183,184]. Therefore, the prevalence of these species that cause the activation of the immune system may elicit anti-tumor immunity in patients who respond to immunotherapy. As a confirmation, studies also suggest that the administration of *Bacteroides*, the sensitization of T cells to LPS, and the modulation of the gut microbiota may revert the resistance to immunotherapy and ameliorate the anti-tumor response [185,186,187]. As reported, gut-derived metabolites such as SCFAs show similar effects and may act on other pathways linked to inflammation through the modulation of intestinal permeability, cell proliferation, apoptosis, and angiogenesis [143,145]. Similarly, the oral administration of indoles may downregulate the inflammatory pathways involved in hepatocarcinogenesis. In CCA, Jin et al. observed that gut microbiota composition may be a prognostic marker of tumor progression, as patients who rapidly progressed showed a higher abundance of phylum *Pseudomonadota* compared with patients with slower tumor progression [186]. In this context, fecal microbiota transplantation (FMT), correcting gut dysbiosis and providing a community of bacteria producing anti-inflammatory substances may restore the integrity of the gut barrier, stimulate the immune system, and rebalance gut microbiota-derived metabolites, favoring the engraftment of *Faecalibacterium* and *Akkermansia spp*. [188,189]. Despite these reports, few studies have analyzed the possible role of FMT on HCC promotion and evolution. However, ongoing studies are evaluating the effects of FMT on HCC therapies. Two clinical trials are evaluating whether FMT from patients with HCC sensitive to Atezolizumab/Bevacizumab may revert a phenotype of resistance to therapy in other patients affected by HCC [190]. A clinical trial of FMT has also been performed in 10 patients with primary sclerosing cholangitis, resulting in the amelioration of intestinal bacterial diversity with a sustained clinical response in the following 24 weeks, with no adverse events being reported. FMT in this contest may reduce the risk of developing CCA, but further data are needed to confirm this hypothesis [191]. Moreover, the use of intestinal microbial strains engineered to metabolize substances such as ammonia or to produce BSH represents a novel strategy to influence liver homeostasis and prevent HCC. These strains are superior compared to wild-type microbes, as they can be directed specifically to the pathogenic molecule or metabolic pathways of interest, sparing the remaining microbial community [192]. Concerning CCA prevention, another strategy may target specific gut bacteria using bacteriophages in patients with PSC, a strategy with reported efficacy in reducing bacterial translocation to bile ducts in mice models [193,194]. All these results are limited by the absence of large randomized clinical trials or meta-analyses on patients. Moreover, the current knowledge of gut microbiota and its involvement in hepatocarcinogenesis is prevalently based on animal models or on samples of fecal microbiota from patients. These models are bidimensional and limited, as they lack the continuous internal and external stimuli that, in real life, influence the metabolism and functions of gut microbiota and its production of metabolites. In the future, the progression of gene sequencing and machine learning-based data analysis will guarantee more reproducible gut microbiota signatures that potentially will also be used as biomarkers.

## 4. Conclusions

In conclusion, evidence suggests a promising role for the gut metabolome as a critical biomarker that must be included in patient evaluation to guarantee the best personalized therapy in the oncological setting. Despite these radiant visions, high costs and the need for multiple platforms and technologies for gut metabolome analysis still limit the diffusion of this analysis in clinical practice. The lack of large randomized clinical trials aimed at the specific analysis of gut-derived metabolites during anti-cancer therapy is another gap in knowledge that should be considered.

## Figures and Tables

**Figure 1 nutrients-16-02372-f001:**
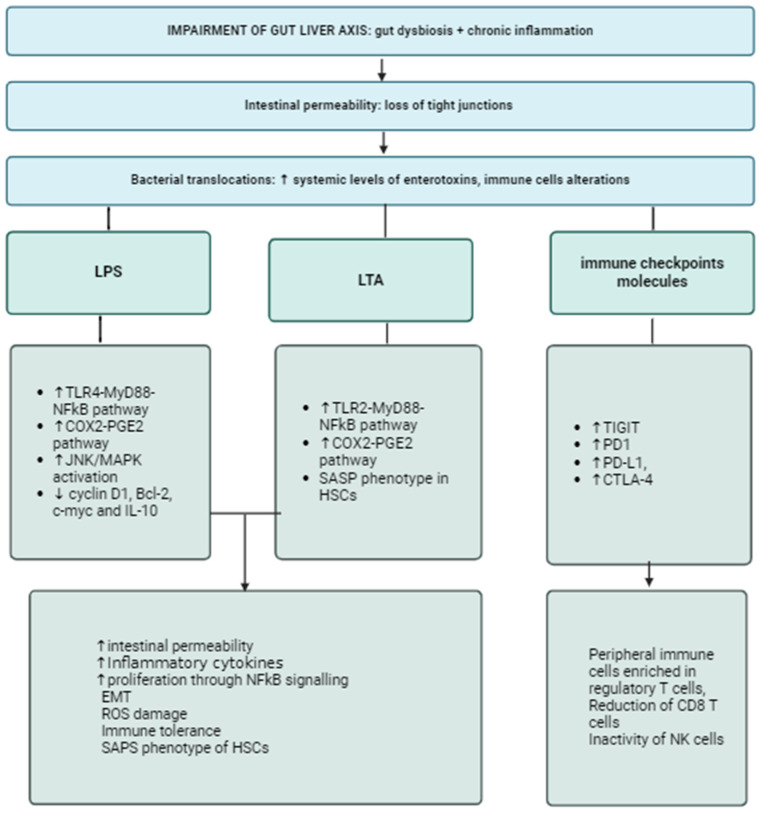
Gut dysbiosis and chronic inflammation leads to increased intestinal permeability, enhancing translocation of enterotoxins, bacteria and their fragments into the bloodstream. LPS and LTA share similar mechanisms, such as stimulating the TLR-MyD88-NFkB pathway, which is a well-known driver of inflammation and cell proliferation, and the activation of the COX2-PGE2 pathway, which causes DNA damage through the production of ROS. LPS also inhibits apoptosis in hepatocytes, enhancing cell replication and favoring the epithelial-to-mesenchymal cell transition. LTA also influences the phenotype of HSCs, leading to their senescence. BT also stimulates the upregulation of immune checkpoint molecules, resulting in increased levels of peripheral anergic cells, such as Tregs, and the loss of CD-8 T cells and NK cells. Abbreviations: LPS, lipopolysaccharide; LTA, lipoteichoic acid; TLR, Toll-like receptor; NFkB, nuclear factor k beta; COX2-PGE2, cyclooxygenase prostaglandin 2; JNK, Janus Kinase; IL, interleukin; SASP, senescence-associated secretory phenotype; HSCs, hepatic stellate cells; TIGIT, T-cell immunoglobulin and immunoreceptor tyrosine-based inhibitory motif; PD1, Programmed Death 1; PD-L1, Programmed Death Ligand 1; CTLA-4, Cytotoxic T-Lymphocyte Antigen 4; ETM, epithelial-to-mesenchymal transition; ROS, reactive oxygen species; Tregs, regulatory T-cells; CD, cluster of differentiation; NK, natural killer.

**Figure 2 nutrients-16-02372-f002:**
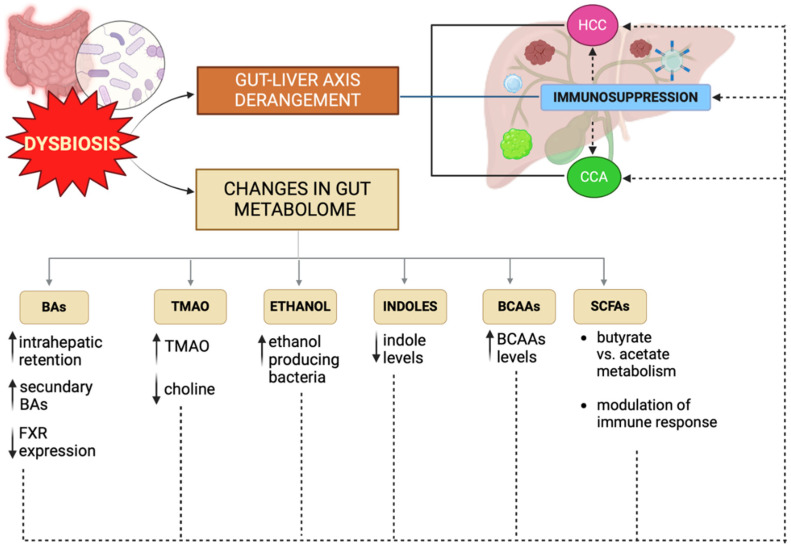
Gut dysbiosis leads to increased intestinal permeability and alterations of the gut–liver axis, leading to liver inflammation. The reiteration of this mechanism promotes an immunosuppressive microenvironment that favors the occurrence of primary liver cancers, including HCC and CCA. Moreover, during dysbiosis, a shift in gut-derived metabolites occurs, deeply influencing tumorigenesis. Dysbiosis is associated with BA retention and increased levels of secondary BAs, especially deoxycholic acid, which drives liver damage, aberrant liver cell proliferation and HCC promotion by the downregulation of FXR expression and the upregulation of the Wnt/β-catenin pathway. In CCA, a decrease in secondary BAs and changes in the glycoconjugated to tauroconjugated BAs ratio is described. High TMAO levels deriving from bacterial choline metabolism are associated with liver cancers, whereas high-choline diets exert a protective effect. Indole derivatives are important for immune cell development and intestinal barrier protection; a reduction in indole levels is typical of chronic liver diseases and high-fat diets. Among SCFAs, butyrate seems to have contradictory effects on HCC, sustaining cancer progression in some studies, and increasing immunotherapy efficacy in others, whereas butyrate seems to reduce cell mitosis in CCA. COnversely, acetate can contrast HCC progression by ILC3 inhibition and interfering with HDAC activity. Ethanol is a well-recognized carcinogenic agent; other than diet, increased ethanol plasma levels can derive from ethanol-producing bacteria, such as *Klebsiella pneumoniae*, which is over-represented in gut dysbiosis. Finally, high BCAA tumor levels and high BCAA intratumor metabolism have been associated with HCC and CCA. Differently, BCAA administration with diet seems to inhibit this process, suggesting a reciprocal relationship between dysbiosis and BCAA metabolism. Abbreviations: HCC: hepatocellular carcinoma; CCA: cholangiocarcinoma; FXR: farnesoid X receptor; TMAO: trimethylamine N-oxide; SCFAs: short-chain fatty acids; HDACs: histone deacetylases; BCAAs: branched-chain amino acids; ILC3s: type 3 innate lymphoid cells; BAs: bile acids. Created with Biorender.com.

**Table 1 nutrients-16-02372-t001:** Microbiota composition in primary liver cancer.

Study Model	Tumor	Microbiota Composition	Other Features
Observational, fecal sample (healthy vs. cirrhosis vs. early HCC) [61]	early HCC	↑ Actinobacteria, *Klebsiella* and *Haemophilus* (producing LPS) ↓ Ruminococcus, Oscillibacter, Faecalibacterium, Clostridium IV, Coprococcus (butyrate-producing bacteria families)	Fecal microbial diversity was decreased from healthy controls to cirrhosis, but it was increased from cirrhosis to early HCC with cirrhosis
Observational, HBV-related HCC (B-HCC) vs. non-HBV and non-HCV-related HCC (NBNC-HCC) [67]	HCC	*↑Escherichia*, *Shigella*, *Enterococcus* *↓ Faecalibacterium*, *Ruminococcus*, *Ruminoclostridium*	Higher species richness of fecal microbiota of B-HCC vs. others
Observational NAFLD-related cirrhosis and HCC vs. NAFLD-related cirrhosis without HCC vs. healthy controls [63]	HCC	↑ *Bacteroides* and *Ruminococcaceae* *↓ Bifidobacterium*, *Akkermansia*	*Akkermansia* and *Bifidobacterium* were inversely correlated with calprotectin concentration, which, in turn, was associated with humoral and cellular inflammatory markers
Case–control HBV-related HCC tissues vs. chronic hepatitis [69]	HCC	*↑ E*. *coli* *S*. *dysenteriae*	↓ Intratumoral microbial heterogeneity of HCC tissues decreased compared with that of nontumor tissues
Observational CCA vs. HCC vs. liver cirrhosis vs. healthy [71]	CCA	*↑ Lactobacillus*, *Actinomyces*, *Peptostreptococcaceae*, and *Alloscardovia*	↑ α-diversities and β-diversities compared to other groups
Observational, in vitro tumor tissue vs. paracancerous tumor [72]	CCA	*↑ Burkholderiales*, *Pseudomonadales*, *Xanthomonadales*, *Bacillales* and *Clostridiales*	*P*. *fungorum* higher in the paracancerous tissues and negatively correlated with CA19.9

Abbreviations: CCA, cholangiocarcinoma; HBV, hepatitis B virus; HCV, hepatitis C virus; HCC, hepatocellular carcinoma; NAFLD, non-alcoholic fatty liver disease.

**Table 2 nutrients-16-02372-t002:** Gut derived metabolites in hepatocarcinogenesis.

Gut Metabolite	Mechanism of Action	Effects	Reference
Bile Acids	DCA induces SASP phenotype in HSCs DCA and G-CDCA induce endoplasmic reticulum (ER) stress with Ca2+ release and promotion of ROS CA, GCA, LCA and CDCA interact with TRAIL and Fas > PKC/MAPK/NFkB and JAk-STAT3 and the PI3-K CDCA and DCA act on EGFR, on Erg-1/MAPK signaling [80] and PKC/MAPK/NFkB BAs reduce FXR activity through SIRT1 in hepatocytes via the Wnt/β-catenin pathway	fibrogenesis chronic inflammation chronic inflammation, fibrogenesis cell proliferation cell proliferation	[32] [6,25,83] [81,82] [79,80,85] [90]
Choline and TMAo	↑ ROS ↓ intrahepatic triglycerides, resulting in a higher risk of metabolic-associated disease, including HCC [101,102,103] activation of mTOR signaling upregulation POSTN gene ↑ f FXR-antagonists ↑ Wnt/β-catenin pathway	DNA damage cell necrosis cell proliferation	[98,99,100,101] [105,106,107] [109,110,111] [90,114]
SCFAs	GPR41, GPR43 or GPR109/PKC/ERK, PKA/ERK activation DNA epigenetic modifications and HDAC inhibition Butyrate ↑ regulatory T cells ↑ IL-10 by microbiota antigen-specific Th 1 cells ↓macrophages in the lamina propria	cell proliferation DNA damage immune suppression	[125,129] [137] [139] [134]
Ethanol	↑ ADH, NADH, CYP2E1	DNA synthesis and repair mucosal injury and cellular DNA instability cell necrosis	[153,154,162,164]
BCAA	↑ IRS1/PI3K/AKT/mTORC1 ↑ catabolic enzymes of BCAAs in tumor cells, ↑ accumulation of branched-chain ketoacids	cell proliferation, angiogenesis and apoptosis	[167,168,169,170,171,172]
Indoles	↓ indole-3-acetate intestinal levels in HFD ↓ fatty acid oxidation	disruption of the intestinal barrier, overexpression of inflammatory cytokines and inhibition of immune cells	[117,118,119]

Abbreviations: alcohol dehydrogenase (ADH), bile acids (BAs), branched-chain amino acids (BCAAs), chenodeoxycholic acid (CDCA), deoxycholic acid (DCA), endoplasmic reticulum (ER), farnesoid X receptor (FXR), glycocholic acid (GCA), hepatic stellate cells (HSCs), histone deacetylase activity (HDAC), lithocholic acid (LCA), mammalian target of rapamycin (mTOR), nicotinamide adenine dinucleotide (NADH), periostin (POSTN), prostaglandin E2 (PGE2), reactive oxygen species (ROS), senescence-associated secretory phenotype (SASP), short-chain fatty acids (SCFAs), TNF related apoptosis inducing ligand (TRAIL), trimethylamine-N-oxide (TMAo).

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
