# Peer review of "Role of Gut Microbial Metabolites in the Pathogenesis of Primary Liver Cancers"

_nutrients, 2024, doi:10.3390/nu16142372_

Round 1

Reviewer 1 Report

Comments and Suggestions for Authors

The reviewed passage provides a deep insight into the role of gut-derived metabolites in maintaining liver homeostasis, inflammation, and tumorigenesis, particularly in the context of diagnosing and treating primary liver tumors. Specifically, it discusses the application of metagenomics and proteomics techniques in identifying and utilizing these metabolites, highlighting the importance of liquid biopsy in cancer detection.

The authors  address issues related to metabolites, such as the breakdown of branched-chain amino acids (BCAAs) and blood tryptophan levels, which are altered in patients with various types of cancers. Although research on HCC is limited, low levels of butyrate in patients with early HCC have been mentioned, opening the field for further studies.

In conclusion, the authors suggest that the gut metabolome could become a critical biomarker in oncology, enabling a more personalized approach to therapy. However, high costs and the need for advanced technologies present challenges to the widespread application of these analyses in clinical practice. Additionally, the lack of large randomized clinical trials is a gap that needs to be addressed to fully exploit the potential of gut-derived metabolites in cancer treatment.

Author Response

The reviewed passage provides a deep insight into the role of gut-derived metabolites in maintaining liver homeostasis, inflammation, and tumorigenesis, particularly in the context of diagnosing and treating primary liver tumors. Specifically, it discusses the application of metagenomics and proteomics techniques in identifying and utilizing these metabolites, highlighting the importance of liquid biopsy in cancer detection.

The authors  address issues related to metabolites, such as the breakdown of branched-chain amino acids (BCAAs) and blood tryptophan levels, which are altered in patients with various types of cancers. Although research on HCC is limited, low levels of butyrate in patients with early HCC have been mentioned, opening the field for further studies.

In conclusion, the authors suggest that the gut metabolome could become a critical biomarker in oncology, enabling a more personalized approach to therapy. However, high costs and the need for advanced technologies present challenges to the widespread application of these analyses in clinical practice. Additionally, the lack of large randomized clinical trials is a gap that needs to be addressed to fully exploit the potential of gut-derived metabolites in cancer treatment.

Comment: We thank the Reviewer for his/her comments. We are glad fro your appreciation. 

Reviewer 2 Report

Comments and Suggestions for Authors

While I find this article very interesting, there are several crucial issues that need to be addressed before publication.

Please including the following publications in the article, which I believe are relevant to the research topic, would be beneficial.

Bi C, Xiao G, Liu C, Yan J, Chen J, Si W, Zhang J and Liu Z (2021) Molecular Immune Mechanism of Intestinal Microbiota and Their Metabolites in the Occurrence and Development of Liver Cancer. Front. Cell Dev. Biol. 9:702414. doi: 10.3389/fcell.2021.702414

Luo W, Guo S, Zhou Y, Zhao J, Wang M, Sang L, Chang B, Wang B. Hepatocellular Carcinoma: How the Gut Microbiota Contributes to Pathogenesis, Diagnosis, and Therapy. Front Microbiol. 2022 Apr 27;13:873160. doi: 10.3389/fmicb.2022.873160. PMID: 35572649; PMCID: PMC9092458.

Please maintain a uniform editing of the expression’s gram positive bacteria, Gram negative bacteria.

Please notice that when mentioning a bacterial species for the first time, its full name is written out. Afterward, its abbreviation must be used. For example, I refer to Escherichia coli. L:  58, 138-139, 172, 176 etc.

L 38-39: “The gut microbiota is made by the complex collection of various microorganisms 38 (bacteria, archaea, eukarya, viruses and parasites)”. Eukaryotes are fungi and metazoic parasites. Please, rephrase the sentence

L 42-47: In which state? Dysbiosis, homeostasis…?

L 54-57: Please rephrase the sentence. Are all members of Bacteroidetes, Firmicutes, and Proteobacteria Gram-negative bacteria?

The introduction also mentions the oral cavity, which should be excluded. Although it is the start of the digestive system, it represents a completely different ecosystem from the gastrointestinal tract.

L 69: 2? The title?

L 81-82: “a qualitative and quantitative modification of the gut microbiota composition”. Please, rephrase.

2.1. The GUT BILIARY LIVER AXIS: between inflammation and immunosuppression. The text is dense and difficult to understand. Please consider creating an image or table to clarify the given information’s.

2.2. GUT DYSBIOSIS and HEPATOBILIARY CARCINOGENESIS. Please ensure that bacterial genus or species names are italicized when referring to them. However, bacterial families and phyla (Line 317, 574) should not be italicized. Maintain consistency in the formatting of bacterial names throughout the text. Also, what is means Escherichia, Shigella and Enterococcus. Do you mean Escherichia spp……? Same as above mentioned, please consider creating a table to clarify the given information’s.

L 225-226. «The main 225 metabolites involved in carcinogenesis are BAs, SCFAs, ethanol, BCAAs and indole-de- 226 rived products.» Who says that?

In Chapter 3, I find it unrelated to the specific purpose of the article, as it simply mentions some chemical substances released in the gut without providing any explanation. The table – figure 1 is on the verge of being too simplistic. Please devise a structured approach to link these metabolites with the pathogenesis of primary liver cancers, which is the main focus of the article.

A huge question. Is the bile acids metabolite of the intestinal microflora? Because the title of the article is "Role of gut microbial metabolites in the pathogenesis of primary liver cancers".

Please discuss the methodological limitations of the research.

Author Response

We thank the Reviewer for her/his suggestions that will definitely improve the relevance of our manuscript. 

Comment 1: Please including the following publications in the article, which I believe are relevant to the research topic, would be beneficial.

Bi C, Xiao G, Liu C, Yan J, Chen J, Si W, Zhang J and Liu Z (2021) Molecular Immune Mechanism of Intestinal Microbiota and Their Metabolites in the Occurrence and Development of Liver Cancer. Front. Cell Dev. Biol. 9:702414. doi: 10.3389/fcell.2021.702414

Luo W, Guo S, Zhou Y, Zhao J, Wang M, Sang L, Chang B, Wang B. Hepatocellular Carcinoma: How the Gut Microbiota Contributes to Pathogenesis, Diagnosis, and Therapy. Front Microbiol. 2022 Apr 27;13:873160. doi: 10.3389/fmicb.2022.873160. PMID: 35572649; PMCID: PMC9092458.

Response 1: We thank the Reviewer for this suggestions, we added these references in the text.

Comment 2: Please maintain a uniform editing of the expression’s gram positive bacteria, Gram negative bacteria.

Response 2: We corrected the text according to your suggestions. 

Comment 3: Please notice that when mentioning a bacterial species for the first time, its full name is written out. Afterward, its abbreviation must be used. For example, I refer to Escherichia coli. L:  58, 138-139, 172, 176 etc.

Response 3: We corrected the text according to your suggestion

Comment 4: L 38-39: “The gut microbiota is made by the complex collection of various microorganisms 38 (bacteria, archaea, eukarya, viruses and parasites)”. Eukaryotes are fungi and metazoic parasites. Please, rephrase the sentence

Response 4: We rephrase it

Comment 5: L 42-47: In which state? Dysbiosis, homeostasis…?

Response 5: We corrected it

Comment 6: L 54-57: Please rephrase the sentence. Are all members of Bacteroidetes, Firmicutes, and Proteobacteria Gram-negative bacteria?

Response 6: We corrected the text according to your suggestions. 

Comment 7: The introduction also mentions the oral cavity, which should be excluded. Although it is the start of the digestive system, it represents a completely different ecosystem from the gastrointestinal tract.

Response 7: we excluded this part in the new version of the manuscript, accordingly to your suggestions

Comment 8: L 69: 2? The title?

Response 8: It was a mystipe

Comment 9: L 81-82: “a qualitative and quantitative modification of the gut microbiota composition”. Please, rephrase.

Response 9: We corrected it

Comment 10: 2.1. The GUT BILIARY LIVER AXIS: between inflammation and immunosuppression. The text is dense and difficult to understand. Please consider creating an image or table to clarify the given information’s.

Response 10: According to your suggestion, we created a table and a figure to clarify this paragraph

Comment 11: 2.2. GUT DYSBIOSIS and HEPATOBILIARY CARCINOGENESIS. Please ensure that bacterial genus or species names are italicized when referring to them. However, bacterial families and phyla (Line 317, 574) should not be italicized. Maintain consistency in the formatting of bacterial names throughout the text. Also, what is means Escherichia, Shigella and Enterococcus. Do you mean Escherichia spp……? Same as above mentioned, please consider creating a table to clarify the given information’s.

Response 11: We corrected it in the text 

Comment 12: L 225-226. «The main 225 metabolites involved in carcinogenesis are BAs, SCFAs, ethanol, BCAAs and indole-de- 226 rived products.» Who says that?

Response 12: We rephrase this

Comment 13: In Chapter 3, I find it unrelated to the specific purpose of the article, as it simply mentions some chemical substances released in the gut without providing any explanation. The table – figure 1 is on the verge of being too simplistic. Please devise a structured approach to link these metabolites with the pathogenesis of primary liver cancers, which is the main focus of the article.

Response 13: We thank the Reviewer. We maintain the figure but we decide to include  a new table to summarize the role of any class of metabolite in the development of liver cancer. Regarding the text, it was not possibile for us to better clarify the mechanisms that link gut derived metabolites and hepatocarcinogenesis, since its a growing field and we mentioned all the known mechanisms. Our paper wants to highlight the current knowledge that, in some cases such as TMAo or indoles, are still poor.  We add some lines in the chapter of BAs to make the reading more fluent. 

Comment 14: A huge question. Is the bile acids metabolite of the intestinal microflora? Because the title of the article is "Role of gut microbial metabolites in the pathogenesis of primary liver cancers".

Response 14: Since secondary Bile Acids are a product of gut microbiota we think they should be included in this category.

Comment 15: Please discuss the methodological limitations of the research

Response 15: We add a section on the methodological limitations in the discussion

Reviewer 3 Report

Comments and Suggestions for Authors

The authors aim to provide in their review the role of gut microbial metabolites in the pathogenesis of primary liver cancers

Recently, more and more investigation on the microbiome-gut-liver axis enhances our understanding of how gut microbiota promotes liver disease and even HCC development. 10.3390/cancers15194875

 Studies published over the last 10 years, revealed that gut microbiota has an important role in human health. Typically, gut microbiota substantially benefits its host, especially in terms of immunity and metabolism10.1038/nri.2016.42 , but gut microbiota has been increasingly recognized to be related to disease processes 10.1038/nm.4185; 10.3390/microorganisms11112730

Studies investigating the occurrence of HCC in obese mice 10.1038/nature12347 discovered that genetic or dietary obesity caused changes in gut microbiota, thus elevating the content of secondary bile acid deoxycholic acid (DCA), the metabolite identified to induce DNA injury. The authors should describe the causal relation between obesity, nonalcoholic steatohepatitis and HCC, since a  high incidence of HCC  is related to NASH and NAFLD, also the prevalent species  of gut microbiota in obese patients and pathogenetic mechanisms of disease progression  10.3390/biology13040243 -; 10.1136/gutjnl-2019-319664; 10.18632/oncotarget.8466; 10.3390/jcm13020420

Gut microbial alterations may serve as biomarkers of HCC disease since they are related to liver disease development, ranging from cirrhosis/fibrosis to cancer 10.3390/nu10101457; 10.1177/1758835919848184. The authors should mention and add the references.

Comments on the Quality of English Language

Minor English  Language Editing is required

Author Response

comment 1

The authors aim to provide in their review the role of gut microbial metabolites in the pathogenesis of primary liver cancers

Recently, more and more investigation on the microbiome-gut-liver axis enhances our understanding of how gut microbiota promotes liver disease and even HCC development. 10.3390/cancers15194875

 Studies published over the last 10 years, revealed that gut microbiota has an important role in human health. Typically, gut microbiota substantially benefits its host, especially in terms of immunity and metabolism10.1038/nri.2016.42 , but gut microbiota has been increasingly recognized to be related to disease processes 10.1038/nm.4185; 10.3390/microorganisms11112730

Studies investigating the occurrence of HCC in obese mice 10.1038/nature12347 discovered that genetic or dietary obesity caused changes in gut microbiota, thus elevating the content of secondary bile acid deoxycholic acid (DCA), the metabolite identified to induce DNA injury. The authors should describe the causal relation between obesity, nonalcoholic steatohepatitis and HCC, since a  high incidence of HCC  is related to NASH and NAFLD, also the prevalent species  of gut microbiota in obese patients and pathogenetic mechanisms of disease progression  10.3390/biology13040243 -; 10.1136/gutjnl-2019-319664; 10.18632/oncotarget.8466; 10.3390/jcm13020420

Gut microbial alterations may serve as biomarkers of HCC disease since they are related to liver disease development, ranging from cirrhosis/fibrosis to cancer 10.3390/nu10101457; 10.1177/1758835919848184. The authors should mention and add the references.

response 1

We thank the reviewer for its suggestions. We included a part on  the relationship between gut microbiota, NAFLD/NASH and HCC in the text. Also, we added the suggested references.

Round 2

Reviewer 2 Report

Comments and Suggestions for Authors

L. 46-49 etc....bacterial families and phyla  should not be italicized. Please correct it.

I am sorry, butI give up.Comment 3: Please notice that when mentioning a bacterial species for the first time, its full name is written out. Afterward, its abbreviation must be used.

L. 121 & 124. Fusobacterium nucleatum and afterward  F. nucleatum

L. 59 & 137 Escherichia coli and afterward E.coli

Comment 14: A huge question. Is the bile acids metabolite of the intestinal microflora? Because the title of the article is "Role of gut microbial metabolites in the pathogenesis of primary liver cancers".

Response 14: Since secondary Bile Acids are a product of gut microbiota we think they should be included in this category.

Please provide a brief explanation in the text justifying the placement of the bile acids in this chapter (meaning Gut metabolome)

Author Response

Comment nr 1:

L. 46-49 etc....bacterial families and phyla  should not be italicized. Please correct it.

Response nr 1

We correct the italics for bacterial phyla, but  the Reviewe should be aware that families can be written in italics according to CdC guidelines:  https://wwwnc.cdc.gov/eid/page/scientific-nomenclature

Comment nr 2

I am sorry, butI give up.Comment 3: Please notice that when mentioning a bacterial species for the first time, its full name is written out. Afterward, its abbreviation must be used.

L. 121 & 124. Fusobacterium nucleatum and afterward  F. nucleatum

L. 59 & 137 Escherichia coli and afterward E.coli

Response nr 2

We corrected it. 

Comment nr3 

Please provide a brief explanation in the text justifying the placement of the bile acids in this chapter (meaning Gut metabolome)

Response nr 3

We already mentioned in the paper ( L 274-279)

Reviewer 3 Report

Comments and Suggestions for Authors

All queries have been addressed

Author Response

Comment 1 All queries have been addressed

Response 1 We thank the Reviewer again for her/his suggestions.